# Can National Registries Contribute to Predict the Risk of Cancer? The Cancer Risk Assessment Model (CRAM)

**DOI:** 10.3390/cancers14153823

**Published:** 2022-08-06

**Authors:** Dorte E. Jarbøl, Nana Hyldig, Sören Möller, Sonja Wehberg, Sanne Rasmussen, Kirubakaran Balasubramaniam, Peter F. Haastrup, Jens Søndergaard, Katrine H. Rubin

**Affiliations:** 1Research Unit of General Practice, Department of Public Health, University of Southern Denmark, 5000 Odense, Denmark; 2OPEN—Open Patient Data Explorative Network, Odense University Hospital, 5000 Odense, Denmark; 3Research Unit OPEN, Department of Clinical Research, University of Southern Denmark, 5230 Odense, Denmark

**Keywords:** cancer diagnosis, automated risk calculation, prediction models, register data

## Abstract

**Simple Summary:**

Early identification of individuals with an increased risk of cancer is an important challenge. Danish administrative registers may be useful in this respect because they cover the entire population and include comprehensive and consistently coded long-term data. We aimed to develop a predictive model based on Danish administrative registers to facilitate the automated identification of individuals at risk of any type of cancer. In addition to age, almost all the included factors contributed statistically significantly, but also only marginally, to the prediction models, which means that we have not overlooked obvious information available in the register. Future prediction studies should focus on specific cancer types where more precise risk estimations might be expected. It is our ultimate ambition that an effective model can be used at the point of care, integrated into electronic patient record systems to alert physicians of patients at a high risk of cancer.

**Abstract:**

Purpose: To develop a predictive model based on Danish administrative registers to facilitate automated identification of individuals at risk of any type of cancer. Methods: A nationwide register-based cohort study covering all individuals in Denmark aged +20 years. The outcome was all-type cancer during 2017 excluding nonmelanoma skin cancer. Diagnoses, medication, and contact with general practitioners in the exposure period (2007–2016) were considered for the predictive model. We applied backward selection to all variables by logistic regression to develop a risk model for cancer. We applied the models to the validation cohort, calculated the receiver operating characteristic curves, and estimated the corresponding areas under the curve (AUC). Results: The study population consisted of 4.2 million persons; 32,447 (0.76%) were diagnosed with cancer in 2017. We identified 39 predictive risk factors in women and 42 in men, with age above 30 as the strongest predictor for cancer. Testing the model for cancer risk showed modest accuracy, with an AUC of 0.82 (95% CI 0.81–0.82) for men and 0.75 (95% CI 0.74–0.75) for women. Conclusion: We have developed and tested a model for identifying the individual risk of cancer through the use of administrative data. The models need to be further investigated before being applied to clinical practice.

## 1. Introduction

Early identification of individuals at a high risk of cancer is an important challenge for all healthcare systems. In recent years, the focus has been on using healthcare data for risk assessment models in cancer [1,2]. Denmark has a long tradition of collecting comprehensive healthcare data [3,4]. However, these data have yet to be used for cancer prediction tools including all-type cancers to be applied in a clinical setting.

An advantage of register-based tools is that the use of existing administrative data eliminates extensive data collection and recall bias. Such tools lend themselves to automatic risk calculations, which may be useful for the systematic detection of high-risk individuals.

Cancer is a major disease burden due to the high morbidity, mortality, diagnosis, and treatment costs and hence the substantial financial costs [5]. Although cancer is a heterogeneous group of diseases, most cancer types’ prognosis is strongly associated with the stage of disease at diagnosis [6]. Therefore, early diagnosis and treatment is important to improve prognosis. Several initiatives have been launched to facilitate early diagnosis of cancer. Screening programs only exist for a few cancer types and are based on age as a predictive factor for cancer [7]. Hence, the majority of cancer patients must be diagnosed on the basis of symptoms [8]. However, recent studies have shown that the predictive value of cancer specific symptoms is low [8,9,10,11,12]. To increase early-stage diagnosis, other approaches may add to screening and adequate symptom-based referral. Prediction tools based on known risk factors have been developed to estimate the patients’ risk of specific cancers [1,2] with adequate performance at the population level but not at the individual level. However, it is possible to investigate whether models based on an extensive amount of different and highly valid data can be the springboard for effective methods and tools of assessing the risk of having cancer based on Danish healthcare data. Denmark has a large array of high-quality national registers that provide a unique opportunity to perform large population-based studies linking information about diagnoses, medications, etc., at the individual level. Consequently, our overall intention was to test whether the Danish registers were useful for developing a high-precision risk prediction model for quantifying the probability of being diagnosed with cancer and for automated case finding of individuals at risk of cancer.

The objective of the study was to identify the relevant data available in Danish registers for inclusion in a predictive model (the Cancer Risk Assessment Model (CRAM)) for automated case finding of individuals at risk of any type of cancer. Second, the aim was to assess the performance of CRAM in a validation cohort stratified by sex, and, thirdly, to investigate the impact of including socioeconomic status in the model, using the data available from the Danish registers.

## 2. Subjects and Methods

### 2.1. Study Design

This study was a nationwide register-based cohort study using data from the Danish national registers covering all individuals in Denmark aged 20 years or above in 2017 with a 10-year look-back period (2007 to 2016).

### 2.2. Data Sources

In Denmark, all inhabitants are provided with a unique civil registration number (CRN) issued at birth or when immigrating to Denmark, which is used as the key identifier in all health and social registers. The Danish healthcare system is tax-funded and provides equal access to universal healthcare services [13].

Statistics Denmark is a national organization in Denmark that is responsible for collecting statistical information about Danish society. We used data on demographic factors, vital status, employment status, education, and personal income [14,15]. Data on marital status, and ethnicity were extracted for 1 January 2017, whereas income and occupational status data were extracted for the year 2016 to avoid any lay-year “illness effect”. Details of the variables are described in Appendix A.

### 2.3. Study Population

The Danish Civil Registration System (CRS) includes all persons living in Denmark [16] and was used to identify persons for inclusion in the study population. The study population included all individuals aged 20 years or above on 1 January 2017. Persons with a cancer diagnosis (ICD-10 code C0–C9, not counting C44) between 1 January 2007 and 31 December 2016 were excluded (Figure 1). Death or emigration in 2017 did not lead to exclusion.

### 2.4. Outcome (Cancer)

The Danish Cancer Registry (DCR) contains the data of all cases of cancer in the Danish population, including date of diagnosis and tumor characteristics [17]. We used this register to identify all cases of cancer during 2017 (ICD-10 codes: C0–C9) (excluding nonmelanoma skin cancer C44) and any cases of cancer prior to 2017 (exclusion criterion).

### 2.5. Conditions of Interest (Exposure)

The Danish National Patient Register (NPR) [18], The Danish National Prescription Registry (DNPR) [19] and The Danish National Health Service Register (NHSR) [20] were used to retrieve information on exposure variables. The NPR includes all inpatient and outpatient hospital visits, including the main medical reason for diagnostic procedures or treatment. From the NPR, we retrieved information on all ICD-10 codes at Level 3 (both somatic (1607 codes included), psychiatric (1272 codes included), and private hospital contacts (1358 codes included)) given as primary or secondary diagnoses from 2007 to 2016. The DNPR contains individual data on all dispensed prescription pharmaceuticals sold in Danish community pharmacies. We used ATC codes from the DNPR at Level 3, recorded as binary variables (yes/no) in the exposure period (2007–2016). The ATC codes had to be registered at least twice during the exposure period to be recorded as “yes” (89 ATC codes included).

The NHSR contains information about activities in primary healthcare, including all general practitioner (GP) contacts [20]. We obtained information on the number of contacts with GPs and selected practicing specialists, and the procedures and measurements issued by a GP from 2007 to 2016 (310 categories included) (Appendix A).

Coding details on age, sex, marital status, country of origin, income, educational and occupational status, and comorbidity can be found in Appendix A.

### 2.6. Statistical Analysis

The study population characteristics are reported as numbers and frequencies for categorical variables and as means and standard deviations or medians and interquartile ranges for numerical variables. Age on 1 January 2017 was categorized into 5-year categories.

The study population was randomly split into 50% as a development cohort, 25% as a validation cohort, and 25% as a test cohort stratified by group (cancer versus control) and sex. To enable accurate comparisons with competing prediction models, we withheld the test dataset for future analysis. We applied a three-step variable selection procedure, stratified by sex, on the development datasets. In the first step, we excluded conditions and ATC codes that occurred in <0.1% of the development cohort during the exposure period. In the second step, we carried out a backward selection with a p-value cut-off of 0.05 on variables remaining after the first step by logistic regression for cancer in 2017 separately for hospital diagnoses, ATC codes, and number of contacts per year with GPs and specialists. In the third step, we carried out a similar backward selection with a *p*-value cut-off of 0.01 combining the selected conditions/ATC codes/contacts with GPs from the second step and age (in 5-year age groups) (Model A).

To investigate the impact of socioeconomic status (SES) on cancer risk, the third step was repeated, including civil status, income, education level, occupation, and country of origin (Model B). Moreover, we constructed a model including age groups only (Model Age), and a fourth model including SES variables only (Model SES) to determine the predictive power of these aspects on their own.

To evaluate the resulting models, we applied the models to the validation cohort and calculated a receiver operating characteristic (ROC) curve based on the predicted probabilities and estimated the corresponding area under the curve (AUC). For predicted risk strata (0–1%, 1–2%, 2–3%, 3–4%, 4–5%, >5%), we calculated the observed cancer frequencies. Furthermore, we evaluated the prediction models by determining the sensitivity, specificity, positive predictive value (PPV), and negative predictive value (NPV) for 1-year cancer risk cut-offs of 1% as well as 5%.

## 3. Results

Denmark had a total population of 5.7 million individuals in 2017. After excluding persons who did not meet the inclusion criteria, the study population consisted of 4.2 million persons (Figure 1). The characteristics of the study population are shown in Table 1.

### 3.1. Cancer Outcome

In total, 32,447 (0.76%) individuals in the study population were diagnosed with first-time cancer in 2017, of whom 648 (1.99%) individuals were diagnosed with more than one cancer type during 2017. Overall, the highest frequency for first-time cancer was seen for gastrointestinal cancer (23.9%). When stratified by sex, the most frequent type of cancer was breast cancer (28.4%) in women and cancer of the genital organs (26.3%) in men.

### 3.2. Conditions Related to Cancer and Development of the Predictive Model (CRAM)

We identified 39 predictive risk factors in women (Table 2) and 42 in men (Table 3).

Age above 30 was a strong predictor for cancer in both sexes. In total, 11 of the 39 identified risk factors in women and 13 of the 42 in men were associated with a lower risk of cancer. Eight risk factors were consistent across the two sexes, whereas none of the GP services (by year) or practicing specialists were identical for men and women.

The models including SES differed slightly (Appendix A). Early retirement and retirement increased the risk of cancer, while being from a non-Western country was associated with a lower risk of cancer in both sexes. For men, being from a Western country (other than Denmark) or having an income in the middle or highest tertile was associated with a lower risk of cancer.

### 3.3. Validation of CRAM

Validating Model A on the corresponding cohort resulted in an AUC of 0.82 (95% CI 0.81–0.82) for men and 0.75 (95% CI 0.74–0.75) for women (Table 4 and Figure 2).

When comparing the observed and predicted frequency of cancer cases, we found that individuals with a predicted 1-year cancer risk above and below the cut-off of 1% had an observed cancer risk in the predicted low-risk group of 0.22% in men and 0.36% in women compared with an observed risk (PPV) in the high-risk group of 2.14% in men and 1.64% in women (Appendix A). Similarly, for a 5% risk cut-off, we observed a cancer risk of 0.76% in men and 0.75% in women in the low-risk group compared with 3.75% for men and 2.38% for women in the high-risk group (Appendix A). Furthermore, stratifying individuals by predicted cancer risk resulted in well-calibrated agreement between predicted and observed risk, and for predicted risk between 1% and 5%, but with the predicted risk overestimating the observed risk for predicted risk above 5% (Figure 2).

The odds ratio for being diagnosed with any cancer in 2017 was 9.78 (9.05; 10.57) with a 1% cut-off and 4.96 (4.00; 6.10) with a 5% cut-off among men. For women, the model resulted in an odds ratio of 4.49 (4.20; 4.80) with a 1% cut-off and 3.34 (1.65; 6.05) with a 5% cut-off. The predictive performance of Model A appears in Table 5.

Including the socioeconomic details (Model B) only marginally improved the AUC. Models including age alone resulted in an AUC of 0.81 for men and 0.74 for women, while the model only including socioeconomics resulted in AUCs of 0.75 for men and 0.70 for women (Table 4).

## 4. Discussion

This study is the first study based on a vast number of already available Danish register data, with overall cancer as the outcome. In addition to age, almost all the included factors contributed statistically significantly, but also only marginally, to the prediction models, which means that we have not overlooked obvious register-available information. Given the inclusion of overall cancer as the outcome and the large dataset, the identified predictive risk factors may, to some extent, represent random findings. Future prediction studies should therefore focus on specific cancer types, staging, and testing in clinical care.

The ROC showed that it is possible to make moderately precise models; however, ‘all-type cancer’ is a difficult case due to the heterogenous outcome. Omitting SES from the models only weakened the model marginally. This is a sign of how the different factors in the registers are related. Our results show that age is a strong predictor of cancer: the model including age alone had a higher precision than the model based on SES exclusively. The CRAM model predicts the 1-year cancer risk well up to 5% risk, which is a clinically relevant spectrum of risk, as the 1-year risk of cancer in the general population is below 1%. However, given a predicted risk above 5%, the model may overestimate the risk of cancer.

### 4.1. Strengths and Limitations

This is the first study developing and validating a risk prediction model for cancer in a Danish setting following up on a similar prediction model developed for osteoporotic fractures [21]. A main strength is the nationwide design covering the entire Danish population. The study did not require patient recruitment, which ensured the inclusion of the whole population of interest and therefore avoided selection bias [13,22].

We extracted data from administrative registries and thereby reduced the likelihood of information bias. Due to the public national registries, the results are applicable to everyone with access to the healthcare system. The CRAM is consequently transparent, can be used on an individual level according to the information presented in this article, and can be tested in future cohorts.

A limitation to our study is that we were not able to include information about ‘online’ covariates as symptom of presentation, or lifestyle (for instance, drinking, smoking, and dietary habits), as this information is not available from the administrative registries. The results should therefore be perceived as a supplement to the information received during patient presentation in clinical practice.

Overall, the cancer group had a higher median age than the control group. This age difference was expected, as age is a known risk factor for cancer. Hence, matching the case and control group would have limited the generalizability of the results to the general population level.

We did not take time to event during 2017 and the competing risk of death into account. Although this might have influenced the results slightly, we found that the effect of these factors would be limited, as our 1-year outcome period was short, implying a limited risk of death during the period, and time to cancer diagnosis is neither very informative nor clinically relevant.

We used a classical backward selection method instead of, e.g., machine learning algorithms. This relatively simple approach has the drawback that more complicated patterns of risk prediction could have been overlooked by the model. On the other hand, the advantage was a transparent methodology, resulting in a final model which can easily be reported, interpreted, and implemented without any privacy concerns with respect to the development data.

We excluded patients with previous cancer, as secondary cancer was considered to be clinically different from primary cancer and because we expect patients with earlier cancers to be followed closely in clinical practice, and hence they are not relevant for general screening. We included any type of cancer for this first study. However, risk factors may play different roles for different types of cancers, and a model should be investigated in relation to specific cancer types in future studies.

### 4.2. Comparison with theExisting Literature

The British QCancer prediction models are, to some extent, comparable with our study [23]; however, the QCancer algorithms included 11 types of cancer, whereas the outcome in our study was any type of cancer, improving the usefulness when the concern is cancer in general. Further, the QCancer studies included socioeconomic characteristics in terms of Townsend score, which is a four-variable population-based deprivation score for a geographical area, whereas we had individual data for developing the CRAM.

Our findings of AUC 0.82 for men and 0.75 for women are comparable with a review of studies regarding prediction models for lung cancer, where the included studies found an AUC between 0.57 and 0.879 [24].

In a prospective observational study among patients with hematuria from 110 hospitals across 26 countries referred to secondary care, a prediction model for urinary tract cancer showed an AUC of 0.86 (95% confidence interval: 0.85–0.87) [25].

The modest contribution of SES to our prediction model is somewhat in contradiction to the findings of other studies of prediction models for nonmalignant diseases, where SES in terms of education and income was found to improve the accuracy of predicting cardiovascular disease risk [26,27] and diabetes [28]. This might be explained by the circumstance that the large number of health predictors included in our study covered the risk information which otherwise could be obtained from the SES data.

A growing number of studies have aimed to improve the risk stratification of patients with cancer through multimodal data integration. There has been much recent interest in the use of machine learning (ML) and different artificial intelligence algorithms for cancer predictions. Studies comparing ML with classical statistical models for risk prediction have already been published [29,30,31], and although some of the studies have demonstrated a promising path toward improved risk stratification of patients with cancer, the relevance for clinical purposes remains to be proved.

## 5. Conclusions

We have verified that the Danish administrative registers are useful for developing a cancer risk prediction model (CRAM) for identifying individuals at risk of having any cancer. The CRAM showed moderately accuracy in the validation cohort and included 39 and 42 risk factors for cancer for women and men, respectively. In addition to age, almost all the included factors contributed statistically significantly, but also only marginally, to the prediction models. Future prediction studies should focus on specific cancer types, where more precise risk estimations might be expected.

## Figures and Tables

**Figure 1 cancers-14-03823-f001:**
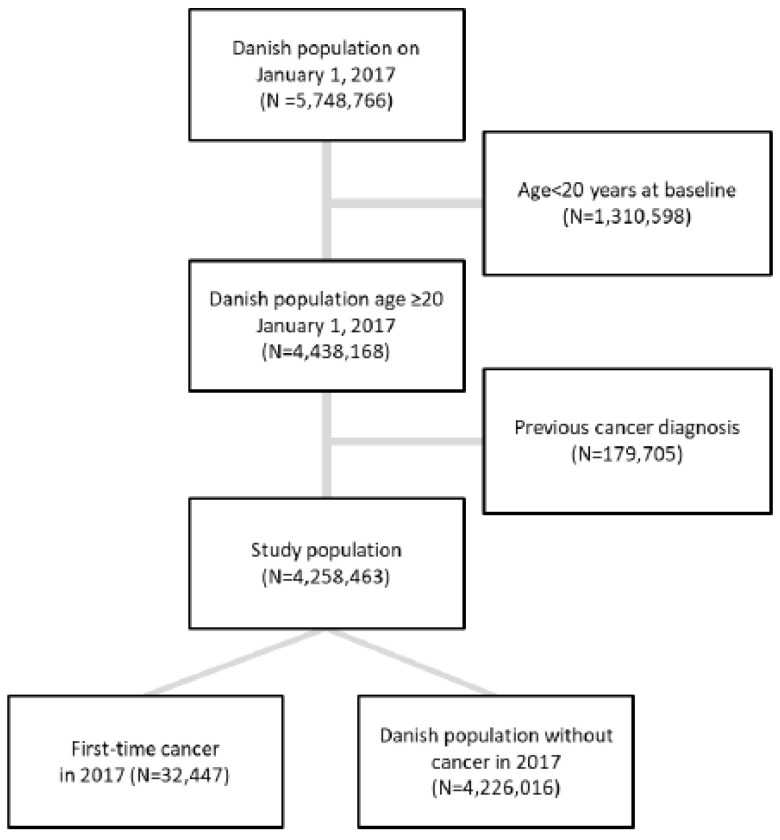
Flow of participants for the CRAM model predictions.

**Figure 2 cancers-14-03823-f002:**
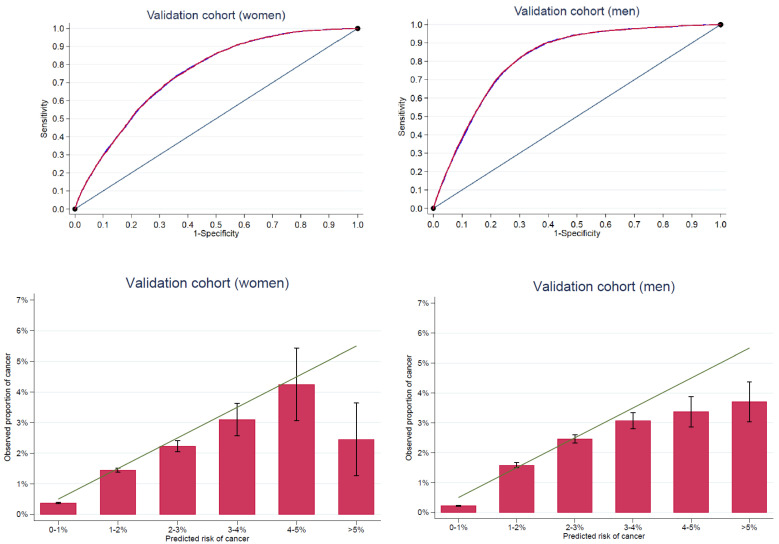
ROC curves in the validation cohort with cancer as the outcome for the CRAM model. Model A (blue) and Model B (red) are almost indistinguishable (**upper figures**). Predicted versus observed cancer cases with 95% confidence bars (**lower figures**).

**Table 1 cancers-14-03823-t001:** Characteristics of the study population stratified by sex, and development and validation cohorts.

	Cases (N (%))					Controls (N (%))			
	Women (*n* = 12,018)		Men (*n* = 12,318)			Women (*n* = 1,604,172)		Men (*n* = 1,565,340)	
	Development (*n* = 7973)	Validation (*n* = 4045)	Development (*n* = 8251)	Validation (*n* = 4067)		Development (*n* = 1,069,726)	Validation (*n* = 534,446)	Development (*n* = 1,043,282)	Validation (*n* = 522,058)
**Age** (Median (Q1–Q3)	67.8 (57.4–75.8)	67.9 (56.8–76.2)	69.7 (61.7–75.8)	69.5 (61.8–75.7)		49.7 (35.1–64.8)	49.7 (35.0–64.8)	48.2 (33.9–62.0)	48.2 (33.9–62.0)
**Age categories**									
Age 20.0–39.9	381 (4.8)	213 (5.3)	220 (2.7)	120 (3.0)		350,901 (32.8)	175,890 (32.9)	364,505 (34.9)	182,329 (34.9)
Age 40.0–59.9	2040 (25.6)	1051 (26.0)	1524 (18.5)	743 (18.3)		375,185 (35.1)	186,510 (34.9)	384,620 (36.9)	192,888 (36.9)
Age 60.0–79.9	4330 (54.3)	2173 (53.7)	5315 (64.4)	2629 (64.6)		275,353 (25.7)	138,047 (25.8)	254,286 (24.4)	126,906 (24.3)
Age ≥80.0	1222 (15.3)	608 (15.0)	1192 (14.4)	575 (14.1)		68,287 (6.4)	33,999 (6.4)	39,871 (3.8)	19,935 (3.8)
**Marital status**									
Married or living with someone	4746 (59.3)	2418 (59.8)	5956 (72.2)	2930 (72.0)		705,074(65.9)	352,347 (65.9)	707,332 (67.8)	353,782 (67.8)
Living alone	3227 (40.5)	1627 (40.2)	2295 (27.8)	1137 (28.0)		364,652 (34.1)	182,099 (34.1)	335,950 (32.2)	168,276 (32.2)
**Ethnicity**									
Danish	7481 (93.8)	3800 (93.9)	7837 (95.0)	3856 (94.8)		929,056 (86.8)	464,024 (86.8)	902,004 (86.5)	451,317 (86.4)
Immigrant	469 (5.9)	233 (5.8)	405 (4.9)	205 (5.0)		126,935 (11.9)	63,477 (11.9)	126,756 (12.1)	63,635 (12.2)
Descendant	23 (0.3)	12 (0.3)	9 (0.1)	6 (0.1)		13,735 (1.3)	6945 (1.3)	14,522 (1.4)	7106 (1.4)
**Country of origin**									
Denmark	7481 (93.8)	3800 (93.9)	7837 (98.3)	3856 (95.3)		929,056 (86.8)	464,024 (86.8)	902,004 (86.5)	451,317 (86.4)
Western countries	264 (3.3)	127 (3.1)	206 (2.6)	111 (2.7)		54,419 (5.1)	27,090 (5.1)	57,819 (5.5)	28,813 (5.5)
Non-Western countries	228 (2.9)	118 (2.9)	208 (2.6)	100 (2.5)		86,238 (8.1)	43,320 (8.1)	83,446 (8.0)	41,923 (8.0)
Unknown or missing	0 (0.0)	0 (0.0)	0 (0.0)	0 (0.0)		13 (0.0)	12 (0.0)	13 (0.0)	5 (0.0)
**Income ^#^**									
High tertile	2106 (26.4)	1038 (25.7)	3018 (36.6)	1521 (37.4)		285,158 (26.7)	143,399 (26.8)	418,974 (40.2)	209,573 (40.1)
Middle tertile	2895 (36.3)	1493 (36.9)	2492 (30.2)	1288 (31.7)		393,342 (36.8)	196,379 (36.7)	311,220 (29.8)	155,220 (29.7)
Low tertile	2972 (37.3)	1514 (37.4)	2741 (33.2)	1258 (30.9)		391,180 (36.6)	194,648 (36.4)	313,038 (30.0)	157,244 (30.1)
Unknown or missing	0 (0.0)	0 (0.0)	0 (0.0)	0 (0.0)		46 (0.0)	20 (0.0)	50 (0.0)	21 (0.0)
**Occupational status**									
Employed	2413 (30.3)	1258 (31.1)	2428 (29.4)	1239 (30.5)		550,737 (51.5)	275,445 (51.5)	635,210 (60.9)	318,155 (60.9)
Unemployed or on welfare payment	64 (0.8)	149 (3.7)	30 (0.4)	119 (2.9)		76,566 (7.2)	40,173 (7.5)	63,911 (6.1)	31,336 (6.0)
Education	292 (3.7)	24 (0.6)	255 (3.1)	15 (0.4)		80,826 (7.6)	38,207 (7.1)	62,085 (6.0)	31,830 (6.1)
Early retirement	744 (9.3)	336 (8.3)	611 (7.4)	316 (7.8)		73,383 (6.9)	36,833 (6.9)	59,022 (5.7)	29,557 (5.7)
Retirement pension	4360 (54.7)	2211 (54.7)	4832 (58.6)	2349 (57.8)		244,640 (22.9)	122,007 (22.8)	183,363 (17.6)	91,385 (17.5)
Unknown or missing	100 (1.3)	67 (1.7)	95 (1.2)	29 (0.7)		43,574 (4.1)	21,781 (4.1)	39,691 (3.8)	19,795 (3.8)
**Education**									
High education	2784 (34.9)	1397 (34.5)	2446 (29.6)	1220 (30.0)		251,820 (23.5)	124,765 (23.3)	238,119 (22.8)	119,297 (22.9)
Medium education	2927 (36.7)	1549 (38.3)	3765 (45.6)	1854 (45.6)		414,171 (38.7)	207,506 (38.8)	472,263 (45.3)	236,730 (45.3)
Low education	2091 (26.2)	1010 (25.0)	1840 (22.3)	899 (22.1)		358,649 (33.5)	179,490 (33.6)	283,304 (27.2)	141,404 (27.1)
Unknown or missing	171 (2.1)	89 (2.2)	200 (2.4)	94 (2.3)		45,086 (4.2)	22,685 (4.2)	49,596 (4.8)	24,627 (4.7)
**Dead in year 2017**	1071 (13.4)	560 (13.8)	1268 (15.4)	620 (15.2)		9135 (0.9)	4501 (0.8)	8427 (0.8)	4325 (0.8)
**Comorbidity**									
Charlson = 0	6500 (81.5)	3307 (81.8)	6588 (79.8)	3292 (80.9)		968,598 (90.5)	484,189 (90.6)	958,628 (91.9)	479,665 (91.9)
Charlson = 1–2	1281 (16.1)	620 (15.3)	1324 (16.0)	609 (15.0)		90,170 (8.4)	44,776 (8.4)	72,649 (7.0)	36,293 (7.0)
Charlson ≥ 3	192 (2.4)	118 (2.9)	339 (4.1)	166 (4.1)		10,958 (1.0)	5481 (1.0)	12,005 (1.2)	6100 (1.2)

N, numbers; %, percent; Q1–Q3, interquartile range; ^#^, tertiles per age category (see the description in the Methods section).

**Table 2 cancers-14-03823-t002:** Model A: Predictive risk factors in the development cohort with cancer as the outcome: women.

Variable	OR (95% CI)	*p*-Value
**Age Categories**		
Age 20–29	Ref	
Age 30–34	2.15 (1.65–2.80)	<0.001
Age 35–39	3.00 (2.35–3.82)	<0.001
Age 40–44	4.67 (3.75–5.81)	<0.001
Age 45–49	6.88 (5.59–8.48)	<0.001
Age 50–54	9.38 (7.66–11.48)	<0.001
Age 55–59	13.00 (10.65–15.87)	<0.001
Age 60–64	17.47 (14.35–21.26)	<0.001
Age 65–69	23.07 (18.99–28.03)	<0.001
Age 70–74	25.81 (21.25–31.35)	<0.001
Age 75–79	28.17 (23.10–34.36)	<0.001
Age 80–84	32.23 (26.32–39.47)	<0.001
Age 85–89	25.44 (20.47–31.61)	<0.001
Age 90–94	18.02 (13.88–23.38)	<0.001
Age 95–99	11.90 (7.56–18.74)	<0.001
Age +100	11.74 (3.71–37.15)	<0.001
**ICD-10 codes**		
T26 (Burns and corrosion confined to the eye and adnexa)	2.43 (1.43–4.13)	0.001
O28 (Abnormal findings on antenatal screening of the mother)	2.06 (1.29–3.29)	0.003
D05 (Carcinoma in situ of the breast)	2.04 (1.52–2.73)	<0.001
E64 (Sequelae of malnutrition and other nutritional deficiencies)	1.77 (1.17–2.66)	0.006
K70 (Alcoholic liver disease)	1.71 (1.22–2.41)	0.002
K13 (Other diseases of the lip and oral mucosa)	1.69 (1.21–2.36)	0.002
J90 (Pleural effusion, not elsewhere classified)	1.64 (1.16–2.33)	0.005
K83 (Other diseases of the biliary tract)	1.62 (1.18–2.23)	0.003
D22 (Melanocytic naevi)	1.52 (1.17–1.97)	0.002
C44 (skin cancer, other type)	1.48 (1.30–1.70)	<0.001
N60 (Benign mammary dysplasia)	1.40 (1.16–1.69)	<0.001
J44 (Other chronic obstructive pulmonary disease)	1.39 (1.26–1.54)	<0.001
I73 (Other peripheral vascular diseases)	1.36 (1.17–1.58)	<0.001
D24 (Benign neoplasm of the breast)	1.36 (1.13–1.65)	0.001
D12 (Benign neoplasm of the colon, rectum, anus and anal canal)	1.31 (1.17–1.46)	<0.001
G40 (Epilepsy)	1.34 (1.09–1.65)	0.006
E04 (Other nontoxic goiter)	1.28 (1.11–1.47)	<0.001
R00 (Abnormalities of the heartbeat)	1.24 (1.06–1.46)	0.009
D25 (Leiomyoma of uterus)	1.24 (1.06–1.47)	0.009
VRK (Perioperative bleeding (ml))	0.76 (0.62–0.92)	0.006
M15 (Polyarthrosis)	0.63 (0.47–0.85)	0.002
F03 (Unspecified dementia)	0.45 (0.29–0.68)	<0.001
G30 (Alzheimer’s disease)	0.37 (0.21–0.65)	0.001
F22 (Persistent delusional disorders)	0.32 (0.14–0.72)	0.006
**ATC codes**		
N07 (Other nervous system drugs)	1.39 (1.26–1.54)	<0.001
L04 (Immunosuppressants)	1.21 (1.07–1.38)	0.004
C08 (Calcium channel blockers)	1.08 (1.02–1.14)	0.008
N02 (Analgesics)	1.07 (1.02–1.13)	0.004
M05 (Drugs for treatment of bone diseases)	0.89 (0.82–0.96)	0.002
A06 (Drugs for constipation)	0.79 (0.71–0.88)	<0.001
**Practicing specialists** **(Year(s) before cancer diagnosis)**		
Child psychiatry (3 years)	46.64 (6.32–344.33)	<0.001
Plastic surgery (1 year)	1.24 (1.06–1.45)	0.008
Internal medicine (4 years)	0.87 (0.80–0.95)	0.002
Psychiatry (5 years)	0.82 (0.72–0.94)	0.003
**GP consultations and procedures** **(Year(s) before cancer diagnosis)**		
GP spirometry (4 years)	1.11 (1.04–1.17)	0.001
GP spirometry (1 year)	1.09 (1.02–1.15)	0.008
GP point-of-care hemoglobin (1 year)	1.04 (1.01–1.06)	0.001
GP consultation (1 year)	1.02 (1.02–1.03)	<0.001
GP consultation (2 years)	0.99 (0.98–0.99)	<0.001
GP point-of-care hemoglobin (6 years)	0.96 (0.93–0.99)	0.007
**Constant (baseline odds)**	0.0005648 (0.0004691–0.0006799)	<0.001

**Table 3 cancers-14-03823-t003:** Model A: Predictive risk factors in the development cohort with cancer as outcome: men.

Variables	OR (95% CI)	*p*-Value
**Age Categories**		
Age 20–29	Ref	
Age 30–34	1.68 (1.19–2.38)	0.003
Age 35–39	2.63 (1.94–3.58)	<0.001
Age 40–44	3.47 (2.61–4.60)	<0.001
Age 45–49	5.84 (4.50–7.57)	<0.001
Age 50–54	10.41 (8.16–13.29)	<0.001
Age 55–59	18.79 (14.81–23.84)	<0.001
Age 60–64	29.98 (23.72–37.90)	<0.001
Age 65–69	42.80 (33.92–54.02)	<0.001
Age 70–74	57.24 (45.37–72.20)	<0.001
Age 75–79	59.19 (46.76–74.92)	<0.001
Age 80–84	64.60 (50.78–82.17)	<0.001
Age 85–89	66.24 (51.47–85.26)	<0.001
Age 90–94	55.49 (41.02–75.07)	<0.001
Age 95–99	51.39 (30.50–86.57)	<0.001
Age +100	78.90 (18.95–328.52)	<0.001
**ICD-10 codes**		
B18 (Chronic viral hepatitis)	2.21 (1.63–3.00)	<0.001
T23 (Burns and corrosion of the wrist and hand)	1.77 (1.20–2.62)	0.004
K83 (Other diseases of the biliary tract)	1.78 (1.29–2.46)	<0.001
R79 (Other abnormal findings of blood chemistry)	1.66 (1.46–1.90)	<0.001
K70 (Alcoholic liver disease)	1.63 (1.27–2.08)	<0.001
F10 (Mental and behavioral disorders due to use of alcohol)	1.51 (1.34–1.70)	<0.001
E04 (Other nontoxic goiter)	1.46 (1.15–1.87)	0.002
T18 (Foreign body in the alimentary tract)	1.45 (1.11–1.89)	0.007
F17 (Mental and behavioral disorders due to use of tobacco)	1.36 (1.16–1.59)	<0.001
R91 (Abnormal findings on diagnostic imaging of the lung)	1.34 (1.17–1.53)	<0.001
D12 (Benign neoplasm of the colon, rectum, anus and anal canal)	1.32 (1.21–1.45)	<0.001
I70 (Atherosclerosis)	1.28 (1.10–1.49)	0.001
T81 (Complications of procedures, not elsewhere classified)	1.20 (1.05–1.37)	0.006
K57 (Diverticular disease of the intestine)	0.85 (0.76–0.95)	0.005
R29 (Other symptoms and signs involving the nervous and musculoskeletal systems)	0.80 (0.70–0.92)	0.002
F00 (Dementia in Alzheimer’s disease)	0.39 (0.23–0.68)	0.001
**ATC codes**		
N07 (Other nervous system drugs)	1.22 (1.10–1.34)	<0.001
R03 (Adrenergics, inhalants)	1.12 (1.05–1.20)	0.001
C08 (Calcium channel blockers)	1.11 (1.05–1.17)	<0.001
N02 (Analgesics)	1.07 (1.02–1.13)	0.004
A12 (Mineral supplements)	0.89 (0.83–0.97)	0.006
R01 (Nasal preparations)	0.86 (0.80–0.93)	<0.001
A06 (Drugs for constipation)	0.84 (0.75–0.94)	0.002
**Practicing specialists** **(Year(s) before cancer diagnosis)**		
Paediatrics (8 years)	1.26 (1.08–1.47)	0.003
Surgery (1 year)	1.17 (1.06–1.28)	0.001
Dermatologist (3 years)	1.05 (1.02–1.07)	<0.001
Ear specialist (1 year)	1.06 (1.02–1.10)	0.001
Ear specialist (4 years)	0.95 (0.91–0.99)	0.009
Radiology Copenhagen (3 years)	0.90 (0.84–0.96)	0.002
**GP contacts or procedures**		
GP spirometry (5 years)	1.21 (1.05–1.38)	0.007
GP spirometry (6 years)	1.10 (1.03–1.17)	0.002
GP Spirometry (2 years)	1.10 (1.03–1.16)	0.002
GP urine examination (1 year)	1.07 (1.04–1.09)	<0.001
GP laboratory test (10 years)	1.06 (1.02–1.10)	0.003
GP blood sample (1 year)	1.04 (1.03–1.05)	<0.001
GP C-reactive protein testing (1 yr)	1.04 (1.02–1.06)	<0.001
GP telephone consultation (1 year)	1.01 (1.00–1.02)	0.002
GP e-mail consultation (1 year)	0.98 (0.97–0.99)	<0.001
GP urine examination (4 years)	0.95 (0.92–0.98)	<0.001
Out of hours services, telephone consultation (2 years)	0.95 (0.91–0.98)	0.006
Out of hours service, consultation (6 years)	0.90 (0.84–0.97)	0.006
GP peak flow (9 years)	0.79 (0.66–0.94)	0.008
**Constant (baseline odds)**	0.0003711 (0.0002961–0.0004651)	<0.001

**Table 4 cancers-14-03823-t004:** Comparison of AUCs obtained for different prediction models.

	Men		Women	
	Development Cohort	Validation Cohort	Development Cohort	Validation Cohort
Model	AUC (95% confidence interval)			
Age	0.81 (0.81–0.82)	0.81 (0.80–0.81)	0.75 (0.74–0.75)	0.74 (0.74–0.75)
SES	0.75 (0.75–0.76)	0.75 (0.74–0.76)	0.70 (0.70–0.71)	0.70 (0.69–0.701)
Model A	0.82 (0.82–0.83)	0.82 (0.81–0.82)	0.76 (0.75–0.76)	0.75 (0.74–0.75)
Model B	0.825 (0.82–0.83)	0.82 (0.81–0.82)	0.76 (0.76–0.77)	0.75 (0.74–0.76)

AUC, area under the curve; SES, socioeconomic status (i.e., civil status, income, education level, occupation, and country of origin). Model A contains ICD-10 codes, ATC codes, and GP and specialist contacts; Model B contains ICD-10 codes, ATC codes, GP and specialist contacts, and SES.

**Table 5 cancers-14-03823-t005:** Absolute predictive performance of Model A.

Absolute Predictive Performance of Model A				
Validation cohort	Model A			
Gender	Men		Women	
Individuals	526,125		526,125	
Total cancer cases	4067		4045	
Risk cut-off	1%	5%	1%	5%
Number of subjects predicted above cutoff	151,668	2550	163,438	439
Cancer cases detected	3247	96	2,77	11
Positive Predictive value	2.1%	3.8%	1.6%	2.4%
Sensitivity	78.8%	2.4%	66.2%	0.3%
Odds ratio	9.78	4.96	4.49	3.34
(95%CI)	(9.05;10.57)	(4.00; 6.10)	(4.20; 4.80)	(1.65; 6.05)

## Data Availability

Data supporting the findings of this study were used under a license granted specifically for the current study and therefore are not publicly available according to the data protection regulations of the Danish Data Protection Agency, Statistics Denmark, and the Danish Health and Medicines Authority.

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
