# Peer review of "Can National Registries Contribute to Predict the Risk of Cancer? The Cancer Risk Assessment Model (CRAM)"

_cancers, 2022, doi:10.3390/cancers14153823_

Round 1

Reviewer 1 Report

Thank you for your valuable work.

The paper is really interesting and well-written.

In this nationwide register-based cohort study, covering all individuals in Denmark aged +20 years, the authors  applied backward selection on all variables by logistic regression to develop a risk model for cancer. The study population consisted of 4.2 million persons, 32,447 (0.76%) were diagnosed with cancer in 2017. The authors identified 39 predictive risk factors in women and 42 in men, with age above 30 as strongest predictor for cancer. Testing the model for cancer risk showed modest accuracy with an AUC of 0.82 (95% CI 0.81-0.82) for men and 0.75 (95% CI 0.74-0.75) for women. Finally, the authors have developed and tested a model for identifying the individual risk of cancer by use of administrative data. The models need to be further investigated before applied to clinical practice.

The paper is well-written, and the study is correctly presented. The paper has its own topicality, and it is interesting to the general audience. 

Tables and figures are correctly labelled.

Therefore, I consider the paper suitable for publication after minor revision.

Author Response

Thank you for the very positive feedback.

We have reduced to 2 significant figures in Table 4. 

Reviewer 2 Report

The authors made the predictive model (Cancer Risk Assessment Model; CRAM) validated on the Receiver Operator Characteristics (ROC). This predictive model is based on various Danish administrative registries. The authors got the opportunity to get all the data from various Danish registries and this is a very resilient cohort sample size for the prediction model development.    

Major comments:

1.     Provide or show the flow chart from participants to CRAM model predictions, and what type of methods were used in the CRAM. 

2.     Compare and briefly show the other studies utilizing the national registry data for the prediction of cancer if available?    

Minor comments:

1.     Is Danish Cancer Registry containing data on drinking, smoking, drug indulgence, and dietary habits?  Are these factors are included or excluded in this model prediction? 

Author Response

Thank you for the opportunity to revise the manuscript according to the reviewers comments.

Major comments

  1. Provide or show the flow chart from participants to CRAM model predictions, and what type of methods were used in the CRAM. 

Answer: Thank you for the opportunity to elaborate. Figure 1 shows the  flow of participants to the CRAM model. We have extended the legend text to Figure 1. 

  1. Compare and briefly show the other studies utilizing the national registry data for the prediction of cancer if available?    

Answer: To our knowledge this study is the first study utilizing the Danish national register data. 

Minor comments:

  1. Is Danish Cancer Registry containing data on drinking, smoking, drug indulgence, and dietary habits?  Are these factors are included or excluded in this model prediction? 

Thank you for the possibility to clarify this.  We did not include data on drinking, smoking, drug indulgence, or dietary habits. The prediction models are based on exposure data from the Danish National Patient Register, The Danish National Prescription Registry, and the Danish National Health Service Register. These registers do unfortunately not include the above-mentioned information. We have elaborated on this in the manuscript. 

The Danish Cancer Registry provided us with data for the outcome measure: all cancer during 2017.